# DQA: An Efficient Method for Deep Quantization of Deep Neural Network Activations

**Wenhao Hu, Paul Henderson, José Cano**
School of Computing Science, University of Glasgow, Scotland, UK

## Abstract

Quantization of Deep Neural Network (DNN) activations is a commonly used technique to reduce compute and memory demands during DNN inference, which can be particularly beneficial on resource-constrained devices. To achieve high accuracy, existing methods for quantizing activations rely on complex mathematical computations or perform extensive searches for the best hyper-parameters. However, these expensive operations are impractical on devices with limited computation capabilities, memory capacities, and energy budgets. Furthermore, many existing methods do not focus on sub-6-bit (or deep) quantization.

To fill these gaps, in this paper we propose DQA (Deep Quantization of DNN Activations), a new method that focuses on sub-6-bit quantization of activations and leverages simple shifting-based operations and Huffman coding to be efficient and achieve high accuracy. We evaluate DQA with 3, 4, and 5-bit quantization levels and three different DNN models for two different tasks, image classification and image segmentation, on two different datasets. DQA shows significantly better accuracy (up to 29.28%) compared to the direct quantization method and the state-of-the-art NoisyQuant for sub-6-bit quantization.

## 1 Introduction

Quantization is a popular compression technique to reduce the compute and memory demands of Deep Neural Networks (DNNs). During inference, DNN weights are typically fixed, so they can be quantized offline. However, activations are dynamically generated, which means that effective quantization methods must dynamically quantize activations online.

There are many methods for quantizing activations. The most straightforward one is to directly use the mathematical definition of quantization [5], but it provides limited accuracy. To get higher accuracy, more sophisticated approaches are used. For example, NoisyQuant [13] injects noise in the activations before quantization and removes the noise after de-quantization (the opposite operation to quantization). However, these methods are not specifically designed for resource-constrained devices, where the computational cost of expensive mathematical operations (e.g. matrix multiplication) or large online search spaces is impractical. Furthermore, many of these methods do not focus on deep quantization (e.g., quantizing to less than 6 bits), which is preferred for devices with limited memory.

To fill these gaps, in this paper we propose DQA (Deep Quantization of DNN Activations), a new method to deeply quantize DNN activations suited for resource-constrained devices that provides high accuracy. DQA first determines the *importance* of each activation channel offline using training or calibration datasets. For important channels, DQA quantizes their values with $m$ extra bits first and then right-shifts the $m$ bits to achieve the target number of bits while storing the shifting errors using Huffman coding. Then, the shifting errors are decoded and added back to the corresponding channels during the de-quantization phase. For unimportant channels, DQA uses a direct method to quantize and de-quantize activations, i.e., it simply applies the mathematical definition of quantization/de-quantization [5; 11]. By learning important channels offline, DQA saves significant computational

Second Workshop on Machine Learning with New Compute Paradigms at NeurIPS 2024 (MLNCP 2024).

resources during DNN inference. In addition, by using $m$ more bits, DQA can quantize important channels with a different number of bits (i.e., mixed-precision quantization [1; 15]), reduce the quantization error exponentially, and efficiently shift back to the target bits without expensive mathematical computations. It is also important to note that: i) by right-shifting, all the quantized values can ultimately be stored using the same bit length, thus avoiding wasted storage [12]; ii) by applying Huffman coding to the shifting errors, DQA can reduce the extra memory overhead.

We evaluated DQA with three sub-6-bit quantization levels (3, 4, and 5 bits) on two different datasets (CIFAR-10 [10], and CityScapes [2]) for three different DNN models (ResNet-32 [7], MobileNetV2 [8] and U-Net [16]) for two different tasks (image classification and image segmentation). Overall, DQA shows significantly better accuracy (up to 29.28%) than two existing methods: direct quantization and the state-of-the-art NoisyQuant [13].

The main contributions of this paper are as follows:

- We propose DQA, a deep (sub-6-bit) quantization method for DNN activations that is especially relevant for resource-constrained devices. DQA deals with important activation channels separately and leverages Huffman coding to optimize memory usage.

- We explore the patterns of shifting errors in DQA to justify our choice of Huffman coding.

- We conduct a detailed evaluation of DQA and compare it with two existing methods, direct quantization and NoisyQuant [13], clearly outperforming them on accuracy.

## 2   Background and Related Work

Quantization is a widely used compression method to lower the precision format of parameters in DNNs, which reduces memory storage and computational requirements [5; 9]. The initial DNN parameters, typically using a floating-point format (e.g., 32 bits), are converted into fixed-point or integer values that require fewer bits (e.g., 16 or 8 bits). The process normally includes clipping and rounding operations. Clipping restricts the minimum and maximum values of the quantization, whereas rounding approximates values to the nearest integers, which causes rounding errors that cannot be recovered during de-quantization. De-quantization is the opposite operation of quantization that approximately recovers the quantized values back to floating point.

Quantization methods are typically divided into uniform and non-uniform. In uniform quantization, the quantized values are evenly spaced, while in non-uniform quantization they are not [5]. Uniform quantization can be further divided into symmetric and asymmetric. In symmetric quantization, the clipping range is symmetric with respect to the origin, while in asymmetric quantization it is not [5]. In this paper, we use uniform symmetric quantization due to its popularity and relatively low cost of computation [5], and we define the direct uniform symmetric quantization and de-quantization as the **Direct** method (note that this is the same as the direct quantization method for unimportant channels mentioned in Section 1).

It is important to note that keeping the same bit length for the quantized values across the whole DNN model is straightforward but can be sub-optimal for accuracy [15]. To improve this, many methods apply mixed-precision quantization which assigns different bit lengths to different parts of a DNN model, such as layers or channels, to satisfy their different precision sensitivities [1; 15]. However, due to the differences in bit length among the quantized values, mixed-precision quantization can cause inefficiencies such as wasted storage [12]. Our DQA method adapts mixed-precision quantization in an effective way (see Section 3).

Furthermore, previous works on DNN quantization apply it to the model weights and/or activations. For weights, AWQ [11] checks the importance of the weight channels for each layer and scales up important channels before quantization to reduce the rounding errors. For activations, quantization is difficult due to its dynamic nature, i.e., the activation values are typically not known before inference. Therefore, to achieve high accuracy, expensive mathematical operations (e.g., matrix multiplication) or large online search spaces to find the best hyper-parameters are required by many methods. For example, NoisyQuant [13] injects noise, obtained by online search within the given search spaces, to the activations before quantization and removes the noise after de-quantization. However, these previous methods do not focus on deep quantization like sub-6-bit quantization, thus being less suitable for resource-constrained devices.

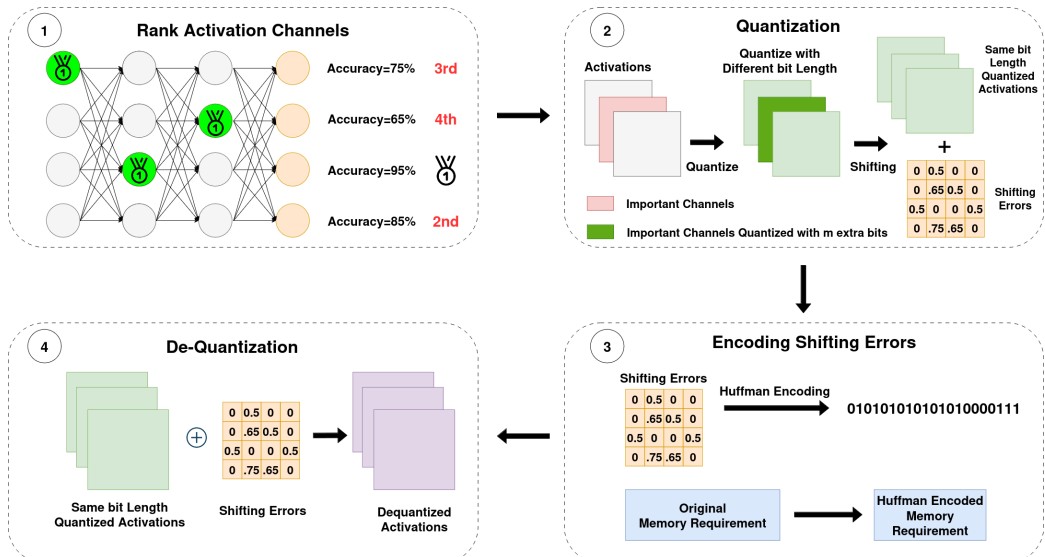

Figure 1: DQA overview. ① offline, rank the activation channels based on importance using training/calibration data and a greedy search algorithm (green circles represent the most important channels for which we skip quantization); ② during inference, quantize important activation channels with $m$ extra bits and then right-shift them while saving the shifting errors; ③ the shifting errors are Huffman-encoded to reduce the memory requirement; ④ de-quantize activation channels. For important channels, decode the Huffman-encoded shifting errors and add them to the quantized activation channel values. For non-important channels, use the direct method to de-quantize.

## 3 Proposed Method: DQA

We now present our proposed DQA method in detail. The main goal of DQA is to efficiently quantize DNN activations, providing high accuracy while minimizing computation and memory requirements, which is especially important on resource-constrained devices. Figure 1 gives an overview of DQA. Given a target of quantization bits $n$:

- ① Offline, DQA uses training/calibration data to rank the activation channels (i.e., their importance) using a greedy search algorithm that quantizes the target activations (see Section 3.1);

- ② During inference, DQA quantizes the important activation channels using $m$ extra bits (i.e., $n + m$ bits in total) while the rest of channels are quantized using $n$ bits. Generally, the higher the accuracy required, the more important channels are needed (we can determine them using a tunable pre-selected ratio of the total channels in each layer). Then DQA right-shifts the important activation channels by $m$ bits and saves the shifting errors. For non-important channels, DQA uses the direct quantization method;

- ③ During inference, DQA encodes the shifting errors for important channels using Huffman coding [17] (see Section 3.2), which reduces the memory requirement.

- ④ During inference, DQA de-quantizates the activation channels. For important channels, it decodes the Huffman-encoded shifting errors and adds them to the quantized activation channel values, thus compensating the information loss. For non-important channels, it uses the direct method to de-quantize.

Algorithms 1 and 2 describe steps ②, ③ and ④ of DQA in more detail for a single layer. Note that channels are quantized/de-quantized one by one (see lines 4 and 3 in Algorithms 1 and 2 respectively); *se* in Algorithm 1 refers to the shifting error. Also note that *Important Channels (I)* are obtained from the rank of the activation channels. Finally, the shifting errors are obtained by reading the lower $m$ bits of each activation value before shifting. Since there are $2^m$ possible combinations of lower $m$ bits that correspond to the shifting errors, they can be converted to decimal floating point values using a pre-computed table (which maps the lower $m$ bits to shifting errors).

**Algorithm 1** DQA Layer Quantization

1: **Input:** Activation Channels $A$, Target bits $n$, Extra bits $m$, Important Channels $I$, Huffman Encoder $H_e(.)$
2: **Output:** Quantized $A$, Encoded Shifting Errors e
3: $e = empty$
4: **for** $a \in A$ **do**
5:     **if** $a \in I$ **then**
6:         $\Delta_{n+m} = \frac{|max(A_{layer})|}{2^{n+m-1}}$
7:         $a_{more} = Round(\frac{a}{\Delta_{n+m}})$
8:         $a, se = RightShift(a_{more}, m)$
9:         $e = e \ || \ H_e(se)$
10:     **else**
11:         $\Delta_n = \frac{|max(A_{layer})|}{2^{n-1}}$
12:         $a = Round(\frac{a}{\Delta_n})$
13: **return** $A, e$

**Algorithm 2** DQA Layer De-Quantization

1: **Input:** Quantized Activation Channels $A_q$, Target bits $n$, Encoded Shifting Errors $e$, Important Channels $I$, Huffman Decoder $H_d(.)$
2: **Output:** De-Quantized A
3: **for** $a \in A_q$ **do**
4:     $\Delta_n = \frac{|max(A_{layer})|}{2^{n-1}}$
5:     **if** $a \in I$ **then**
6:         $a = \Delta_n \cdot (a + H_d(e))$
7:     **else**
8:         $a = \Delta_n \cdot a$
9: **return** $A$

## 3.1 Ranking Important Activation Channels Using Greedy Search

We hypothesize that there are important activation channels that can be treated differently for quantization, similar to how AWQ [11] processes weights. We define important activation channels as those for which skipping quantization can yield better accuracy in DNN inference. Since activations are dynamically generated, it would be too computationally expensive to rank activation channels during inference. As a result, we compute the ranks offline using training/calibration data. Assuming that both training/calibration and inference data are drawn from sufficiently similar distributions, the ranks computed offline can be reused during inference. Therefore, we select important channels (which are fixed and based on a tunable pre-selected ratio of the total channels in each layer) offline. Then, the memory requirement can be accurately calculated offline before inference.

To calculate the rank of the activation channels, we avoid brute force and dynamic programming approaches due to their impractical time complexity. Instead, we use greedy search that only requires $O(LN)$ operations, where $N$ is the number of channels and $L$ is the number of layers. In our greedy search algorithm (see Algorithm 3 in Appendix), we iterate the activation channels for each layer. In each iteration, we skip the current activation channel and only quantize the remaining activation channels of the layer. We then run inference on evaluation data to measure the impact of skipping the current activation channel on accuracy. The skipped activation channels that provide the highest accuracy are considered the most important ones for that layer. From layer two onwards, we first quantize the activation channels of the previous layers using their layer ranks to skip the most important channel of each layer.

## 3.2 Quantizing Important Activation Channels

To preserve more information for the important activation channels, and thus obtain higher accuracy, we quantize them with $m$ extra bits, where $m$ is less than or equal to the target number of bits $n$, so that they are quantized using $n + m$ bits. Then, we right-shift the activation values of the important channels by $m$ bits to reach the target bit length $n$. Note that this operation allows storing all quantized values with same bit length $n$, which helps avoid wasted storage [12]. Also note that right-shifting is a computationally cheap operation [4]. Since right-shifting will lead to information loss for the important activation values, we save the shifting errors and add them back during the de-quantization phase to compensate the information loss.

### 3.2.1 Quantization Error

We now formally analyze the shifting errors for the important activation channels. We denote by $I$ the important activation channels. The most straightforward quantization/de-quantization approach for the important activation channels can be expressed as [11]:

$$Q(I) = \Delta_N \cdot Round(\frac{I}{\Delta_N}), \quad where \quad \Delta_N = \frac{|max(I)|}{2^{N-1}} \tag{1}$$

The quantization error of this method is ($re$ is the Rounding Error):

$$Error = |I - Q(I)| = |I - \Delta_N \cdot (\frac{I}{\Delta_N} + re)| = \Delta_N \cdot re \tag{2}$$

The average $re$ is 0.25 [11]. With DQA, $I$ will be quantized with $m$ extra bits first, and then right-shifted also with $m$ bits. Then, the de-quantized important activation channels can be expressed as ($se$ is the Shifting Error):

$$Q(I) = \Delta_N \cdot (RightShift(Round(\frac{I}{\Delta_{N+m}})) + se) \tag{3}$$

Finally, the quantization error becomes:

$$Error = |I - \Delta_N \cdot (\frac{1}{2^m} \cdot (\frac{I}{\Delta_{N+m}} + re) - se + se)| = \Delta_N \cdot \frac{1}{2^m} \cdot re \tag{4}$$

From Equations 2 and 4, it is clear that this approach exponentially ($2^m$) reduces the quantization error. This is expected since we add $m$ more bits of information.

### 3.2.2 Memory Overhead

DQA improves the accuracy by adding $m$ extra bits and saving the shifting errors, which involves some memory overhead. To reduce this overhead, we empirically explore the pattern of the shifting errors. As an example, we study the average frequency distribution of the shifting errors (i.e., the average value of the shifting errors frequencies calculated during inference with each batch of input data) for ResNet-32 and the CIFAR-10 dataset with 3, 4, and 5 bits quantization and $m = 3$.

As we can see in Figure 2, the distributions of shifting errors are not uniform. Therefore, we can use Huffman coding [17] to compress the errors by encoding the most frequent errors with fewer bits, and the less frequent errors with more bits. Note that in general Huffman coding can generate shorter encoding information compared to directly storing $m$ bits, which justifies its use to compress shifting errors.

Note that in our experiments in Section 4, Huffman coding shows benefits in compressing the shifting errors. Specifically, the average compression ratio ($\frac{\text{Size\_of\_Original\_Shifting\_Errors}}{\text{Size\_of\_Huffman\_Code}}$) for Huffman coding on the CIFAR-10 dataset is up to 1.12.

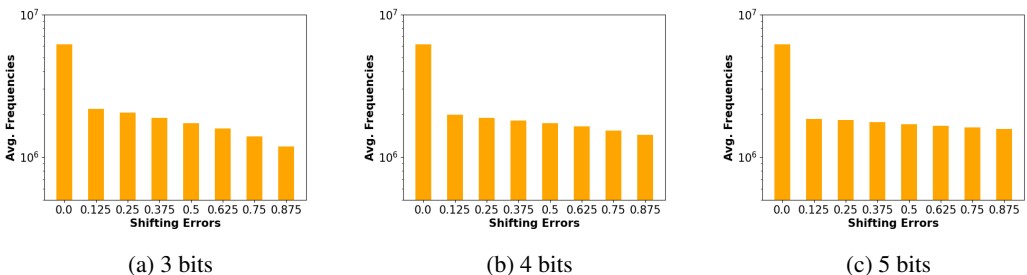

(a) 3 bits          (b) 4 bits          (c) 5 bits

Figure 2: Average frequency distribution of shifting errors for ResNet-32 and CIFAR-10 with 3, 4, and 5 bits quantization and $m = 3$.

## 4 Evaluation

### 4.1 Experimental Setup

For image classification, we use ResNet-32 [7] and MobileNetV2 [8] on the CIFAR-10 dataset [10]. For image segmentation, we use U-Net [16] on the CityScapes [2] dataset. Note that ResNet-32 and MobileNetV2 store the activations of the shortcut connections, whereas U-Net stores the activations of encoder layers. The three DNN models are typical examples where activations need to be i) stored for a relatively long term and ii) compressed to have a manageable peak memory usage.

Table 1: Accuracy (Acc) results for the three DNN models under study.

| (a) Classification: ResNet-32 Original Acc: 92.49% | | | (b) Classification: MobileNetV2 Original Acc: 91.42% | | | (c) Segmentation: U-Net Original Acc: 92.5% | | |
|---|---|---|---|---|---|---|---|---|
| **Method** | **Bits** | **Acc (%)** | **Method** | **Bits** | **Acc (%)** | **Method** | **Bits** | **Acc (%)** |
| Direct | 3 | 52.85 | Direct | 3 | 82.75 | Direct | 3 | 84.41 |
| NoisyQuant | 3 | 71.22 | NoisyQuant | 3 | 87.67 | NoisyQuant | 3 | **92.16** |
| DQA($m = 3$) | 3 | **82.13** | DQA($m = 3$) | 3 | **88.63** | DQA($m = 3$) | 3 | 90.38 |
| Direct | 4 | 87.74 | Direct | 4 | 90.01 | Direct | 4 | 91.54 |
| NoisyQuant | 4 | 89.66 | NoisyQuant | 4 | 90.56 | NoisyQuant | 4 | 91.13 |
| DQA($m = 3$) | 4 | **90.55** | DQA($m = 3$) | 4 | **90.84** | DQA($m = 3$) | 4 | **92.03** |
| Direct | 5 | 91.33 | Direct | 5 | 91.19 | Direct | 5 | **91.48** |
| NoisyQuant | 5 | 91.86 | NoisyQuant | 5 | **91.33** | NoisyQuant | 5 | 91.41 |
| DQA($m = 3$) | 5 | **92.12** | DQA($m = 3$) | 5 | 91.28 | DQA($m = 3$) | 5 | 91.46 |

Since DQA is designed for deep quantization of DNN activations, we set the target quantization levels to 3, 4, and 5 bits; note that $m = 3$ in all cases to simplify the evaluation (we leave as future work the exploration of different values of $m$ for different quantization levels).

We implement our experiments using PyTorch [14] and run them on an Nvidia RTX 3090 GPU, as the main goal of this work is to evaluate accuracy (we leave as future work the exploration of more resource-constrained devices). Every experiment is run 5 times for each quantization level, and we take the average value in each case.

For each DNN model, we created a rank table (which maps the name and activation channel ranks of each layer) for each run of the experiments with a random subset of the training data using greedy search (see Section 3.1). The size of the random training data subset for creating each rank map is 5000 for image classification and 50 for image segmentation. We select important channels with ratios of 40% for image classification and 50% for image segmentation, for all channels in all layers. The batch size is 128 for image classification and 4 for image segmentation in all corresponding experiments, including the direct quantization method and NoisyQuant.

## 4.2 Results

We compare DQA with two methods i) direct quantization (defined in Section 2) and the state-of-the-art NoisyQuant [13]. Even though NoisyQuant was designed for Vision Transformers [3], our experiments show that it also works well for other types of DNNs. Note that all experiments only quantize activations, i.e., without considering weights.

Table 1 shows the accuracy results for the three DNN models under study. Overall, DQA works better for classification tasks than segmentation task. More specifically, for image classification models DQA achieves up to 29.28% higher accuracy than the direct method and NoisyQuant, where ResNet-32 always obtains higher differences than MobileNetV2. We also see that the lower the quantization bits, the more accuracy advantages DQA can provide. For U-Net (image segmentation), we observe that DQA provides higher accuracy (up to 0.9%) than the direct method and NoisyQuant when quantizing with 4 bits, and similar accuracy when quantizing with 5 bits. For 3 bits, DQA improves over direct quantization but NoisyQuant provides the best accuracy value.

## 5 Conclusion

In this paper we proposed DQA, an efficient method that applies deep quantization (with less than 6 bits) to DNN activations and provides high accuracy while being suitable for resource-constrained devices. We evaluate DQA on three DNN models for both image classification and image segmentation tasks, showing up to 29.28% accuracy improvement compared to direct quantization and the state-of-the-art NoisyQuant. As future work, we plan to co-design new versions of DQA and hardware accelerators [6], which will also allow us to evaluate the system performance (such as inference latency) on resource-constrained devices, thus further exploiting the benefits of DQA.

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

# Appendix

---

**Algorithm 3** Ranking Important Activation Channels Using Greedy Search

---

1: **Input:** Model $M$, Training or Calibration Dataset $D$
2: **Output:** All ranks of activation channels $R$
3: $R = \{\}$
4: $P = \{\}$      // Most important activation channels of previous layers
5: **for** $layer \in M$ **do**     // In forward direction
6:     $rank = \{\}$
7:     $highest\_accuracy = 0$
8:     $most\_important\_channel = none$
9:     **for** $channel \in layer.activation\_channels$ **do**
10:         // $IQ$ quantizes activations of previous layers and activations of current layer
11:         $accuracy = IQ(M, layer, channel, P, D)$
12:         $rank = rank \cup (channel, accuracy)$
13:         **if** $accuracy > highest\_accuracy$ **then**
14:             $highest\_accuracy = accuracy$
15:             $most\_important\_channel = channel$
16:     $P[layer] = most\_important\_channel$
17:     $rank = sort(rank)$     // sort by accuracy
18:     $R[layer] = rank$
19: **return** $R$

---

