# OpenReview forum: "DQA: An Efficient Method for Deep Quantization of Deep Neural Network Activations"
_NeurIPS.cc/2024/Workshop/MLNCP — MLNCP Poster_

### Official Review · Reviewer_iz1h · 2024-10-02
**DQA presents a simple yet efficient way to quantize activations to sub-6bit representations while maintaining high performance**

**Rating:** 7
**Confidence:** 4

**Review:**

This is a nice paper. DQA determines the importance of each activation, and depending on the importance, will handle the quantization differently. I've seen lots of works in quantizing weights, but this one tackles the challenging problem of quantizing activations. This is usually difficult since the outputs are not usually known before inference. DQA explains their methodology well. They follow with a number of experiments supporting their claims and how their method is simple and performs better than NoisyQuant, their most related work.

A few minor questions:
- When storing the shift errors using Huffman coding, what is the effective number of bits per value? This might be a nice number to include. I see the authors mention that the overhead is <2% of the memory for model weights but thinking about it in terms of number of effective bits helps for comparison. How does storing the full activation value using Huffman coding compare?
- when trying other values of m, did you see any trends as you increased or decreased m? How should researchers choose m?
- in the table 1a-c, would you be able to add in the accuracies for the original models with no quantization? It would be good to know what the cap is.
- a little typo in figure 2's caption and line 207.


Overall, very nice.

---

### Official Review · Reviewer_ZCcT · 2024-10-06
**A "dual-mode" quantization scheme that still needs some investigation**

**Rating:** 7
**Confidence:** 5

**Review:**

# Summary
The paper presents a novel quantization approach for activation that leverages additional encoded biases for "important" channels. The method exploits sub-6-bit quantization, is tested on three CV tasks, and achieves better accuracy than standard quantization methods.

The paper is generally promising, but it would benefit from a deep evaluation of the memory/accuracy tradeoff.

# Pros
- The paper topic is engaging.
- Method seems general and could potentially be used in many contexts.
- Preliminary results show the effectiveness of the method.

# Cons
- Memory overheads/accuracy tradeoff is not discussed in depth.
- Channel selection tradeoff (how many channels) and costs (in terms of time) are not discussed.
- Execution time overheads for runtime quantization and de-quantization are not discussed.

# Comments
The paper presents an interesting hybrid solution for low-bit quantization.
The paper, at this stage, could be strongly improved by a more detailed evaluation of the accuracy/memory tradeoff enabled by the proposed method:
- First, channel selection and the number of channels selected are important because they impact both the memory overheads and the accuracy of your method. An ablation study would be very appreciated.
- Second, memory overheads and accuracy with respect to the FP32 model should be added in Table 1.
- Third, the additional additive operations at runtime for quantization and de-quantization must be analyzed using realistic devices.

Your background and related work should also consider mixed quantization methods because your quantization scheme ultimately results in a channel-wise dual quantization process.

---

### Official Review · Reviewer_4JXh · 2024-10-07

**Rating:** 3
**Confidence:** 4

**Review:**

Paper Content Summary : the paper claims deep quantization (i.e. sub 6-bit quantization) of DNN (i.e. ResNet, MobileNet and U-Net) is lacking in the current state, where the only few exisiting solutions requires expensive process on device to make the quantized model accurate. Then the paper proposes a process that 1) identifies important channels in activations 2) quantize those with higher precision (n+m bits) 3) right shift to lower the precision to match target compression precision (n bits) 3) bookkeep the right shift errors and compress them using huffman coding.

Reasons to Reject:

First of all, the paper is hard to parse, with lots of repetitive content in abstract, intro, proposed method, result and conclusion. This makes the paper lack depth. I don’t feel like learning a lot or get inspired after reading the paper. But I do think with enough rewriting this can be improved.

Secondly, it is not convincing that the method proposed is superior or novel than existing methods. The paper states that few works compress DNN to sub-6bit, but this is not true [1][2]. The paper studies an outdated problem and the solution proposed is not significantly better in terms of result or conceptual insights.

- In terms of results, the accuracy was claimed to be up to 29% better than others, however the result table (Table 1) shows that this only applies to a single case among the 9 settings tested; for other cases the improvement is trivial or even worse than baselines.
- Conceptually, the method is overly complicated, leading to potential slowdown in inference. For example the different processing of channels disables parallelized matrix operation of quantization. Additional huffman coding and decoding steps are costly too. In addition, the compression rate of huffman coding is shown to be only 1.1 in the paper - leaving the "important" channels essentially taking "m-bits" more than the target n-bit precision. All these overhead defeat the purpose of deep quantization.

Suggestions to Improve the Paper:

- back up the problem statement and motivation, as the problem and the model scope in current paper is studied by many other works already. show the reason why your work has impact.
- address the questions on efficiency and inefficiencies of your algorithm. in various places in the paper, you tried to claim the method is more efficient - but no quantification or evidence to support it. (how expensive online search is? how much more efficient is your approach compared to, say, NoisyQuant? why would you compare the memory consumption to weights? if activation tensor is small to start with, in batch size 1 setting, why bother quantizing it?)
- In terms of writing, I'd suggest keeping abstract and introduction high level, leaving all the algorithm details to specific sections. The first few sections should well support the motivation and impact of this work and leaving the "how to" later. It was very confusing when I read about method details with all the custom terms in abstract and intro without seeing Figure 1 first. Figure 1 was very helpful though.

(final note: I spent a long time parsing the paper trying to find reasons to accept it, and not be rejecting due to writing quality, but this is all I got so far, I can't figure out enough contributions by the paper to support accepting. I hope these revise suggestions could help, now or in the long run.)

[1] Abdolrashidi, AmirAli, et al. "Pareto-optimal quantized resnet is mostly 4-bit." *Proceedings of the IEEE/CVF Conference on Computer Vision and Pattern Recognition*. 2021

[2] Yang, Bin, et al. "2-bit model compression of deep convolutional neural network on ASIC engine for image retrieval." *arXiv preprint arXiv:1905.03362* (2019).

---

### Decision · Program_Chairs · 2024-10-10

Accept (Poster)